# The Pharmacokinetic and Absolute Bioavailability of Cyclosporine (Atopica for Cats^®^) in Cats

**DOI:** 10.3390/vetsci10060399

**Published:** 2023-06-18

**Authors:** Jingyuan Kong, Yuxin Yang, Yu Liu, Yuying Cao, Jicheng Qiu, Pan Sun, Xingyuan Cao

**Affiliations:** 1Department of Veterinary Pharmacology and Toxicology, College of Veterinary Medicine, China Agricultural University, Beijing 100193, China; jingyuankong@outlook.com (J.K.); yuxinyang1314@outlook.com (Y.Y.); ly2226185804@163.com (Y.L.); cyy@cau.edu.cn (Y.C.); 13612020859@163.com (J.Q.); sunpan@cau.edu.cn (P.S.); 2Key Laboratory of Detection for Veterinary Drug Residues and Illegal Additives, Ministry of Agriculture and Rural Affairs of the People’s Republic of China, Beijing 100193, China

**Keywords:** cyclosporine, pharmacokinetic, bioavailability, cats, UPLC-MS/MS

## Abstract

**Simple Summary:**

This study aimed to evaluate the rate and amount of cyclosporine A in cats following oral administration using intravenous administration as a reference. Twenty-four healthy cats were selected and randomly divided into four groups in this study. All animals were adapted in a convenient environment during the whole experiment and no adverse effect was observed. Each animal was given a respective dose only once. Blood samples were collected at scheduled time points and analyzed by the established method. The pharmacokinetic parameters were calculated by professional software used in the whole pharma industry of drug development. As a result, the amount of cyclosporine A absorbed in vivo is from one-fifth to one-third, demonstrating that the medium dose has great potential to get into the body. Due to its common side effects always being associated with the amount of cyclosporine in the blood, it is necessary to monitor the concentration of cyclosporine clinically. Practically, doctors draw blood before or 2 h after taking medicine. In this study, we concluded that taking a blood sample after 4 h has more power to estimate the extent of absorption and will be better at preventing the occurrence of side effects in clinical use.

**Abstract:**

This study aimed to evaluate the absolute bioavailability of cyclosporine in cats by investigating the pharmacokinetic profile after intravenous and oral administration, respectively. Twenty-four clinically healthy cats were enrolled in this study and randomly divided into four groups, namely the intravenous group (3 mg/kg), low oral group (3.5 mg/kg), medium oral group (7 mg/kg), and high oral group (14 mg/kg). Whole blood was obtained at the scheduled time points after a single dose administration and cyclosporine was determined using ultra-performance liquid chromatography–tandem mass spectrometry technology (UPLC-MS/MS). Pharmacokinetic parameters were calculated using the WinNonlin 8.3.4 software via compartmental and non-compartmental models. As a result, the bioavailability values for the low, medium, and high oral groups were 14.64%, 36.98%, and 13.53%, respectively. The nonlinear pharmacokinetic profile was observed in the range from 3.5 mg/kg to 14 mg/kg in cats following oral administration. Whole blood concentrations taken 4 h after oral administration were better correlated with the area under the blood concentration–time curve AUC_0–24_ with a high regression coefficient (R^2^ = 0.896). This concentration would be a greater predictor in the following therapeutic drug monitoring. No adverse effect was observed in the whole study process.

## 1. Background

Cyclosporine, [(3*S*,6*S*,9*S*,12*R*,15*S*,18*S*,21*S*,24*S*,30*S*,33*S*)-30-ethyl-33-[(*E*,1*R*,2*R*)-1-hydroxy-2-methylhex-4-enyl]-1,4,7,10,12,15,19,25,28-nonamethyl-6,9,18,24-tetrakis(2-methylpropyl)-3,21-di(propan-2-yl)-1,4,7,10,13,16,19,22,25,28,31-undecazacyclotritriacontane-2,5,8,11,14,17,20,23,26,29,32-undecone, (C_62_H_111_N_11_O_12_)], isolated from *Tolypocladium inflatum gams*, is a cyclic undecapeptide. As a suppression immune system agent, it has been used in transplant patients, and for the treatment of dermatitis and psoriasis [1,2,3,4].

Cyclosporine depresses the proliferation of cytotoxic T cells by decreasing the lymphocyte expression of the nuclear factor of activated T cell-regulated cytokines [5,6], and by decreasing the antibody production of helper T-cell-dependent B cells, mast cells, basophils and eosinophils, which were involved in allergic and inflammatory actions [1,4,7]. It blocks cyclophilins, acting as peptidyl-prolyl *cis-trans* isomerase, which is very important in the selective inhibition of cytokine–gene transcription [8]. It regulates the immune system by targeting calcineurin-dependent transcription factors to decrease the involved cytokines, such as interleukin 2, interleukin 3, interleukin 4, and interferon-gamma. These cytokines play a heavy role in activating T cells [1,8]. It also affects the production of tumor necrosis factor-alpha and major histocompatibility complex class II [5]. These direct and indirect effects involving the cytokine network may account for the inhibition of cyclosporine in regulating many cell types, cytokines, and receptors.

Cyclosporine has been used to treat autoimmune conditions such as rheumatoid arthritis, lupus erythematosus, and inflammatory bowel disease in humans. In addition, cyclosporine has been used in the treatment of certain types of cancer in people [9,10]. Following oral administration in humans, the traditional cyclosporine formulation, Sandimmune, was variable from patient to patient in a pharmacokinetic study due to its significant molecular weight, being highly hydrophobic, and the site of absorption [11]. The microemulsion formulation, Neoral, improved its bioavailability and reduced the intra-subject discrepancy. Since dogs and cats share many anatomical and physiological similarities with humans, these drugs can also be used in these animals. Due to its highly lipid solubility, it accumulates in the liver and other tissues. It has been used in veterinary medicine for many years for curing some diseases, such as rheumatoid arthritis, psoriasis, Crohn’s disease, nephrotic syndrome, sebaceous adenitis, pemphigus foliaceous, inflammatory bowel disease, anal furunculosis, and myasthenia gravis [10,12]. However, the topical formulation has not been applied practically owing to its poor penetration through the stratum corneum, which forms a solid barrier to the transport of cyclosporine [2]. It is important to note that there is often a poor correlation between the clinical response observed in people and animals treated with the same drug. This may be due to differences in the pharmacokinetic and pharmacodynamic properties of these agents between species. It is therefore necessary to conduct pharmacokinetic studies to determine the optimal dose of the drug that will elicit the maximum therapeutic effect with minimal side effects.

The pharmacokinetics of cyclosporine have been studied extensively in dogs, but little is known about its pharmacokinetics in cats [3]. In recent years, liquid chromatography–tandem mass spectrometry (LC-MS/MS) has replaced immunoassays with its high specificity and low cross-reactivity applied in different matrixes such as whole blood, plasma, and urine [3]. Atopica for Cats^®^ used in this study is an oral formulation of cyclosporine, which was developed in a microemulsion formulation by Elanco US Inc. Greenfield and used in cats only [13], and this formulation has an advantage in that it analyzes the pharmacokinetics of cyclosporine without concerning the biliary action, enzymes, or small intestinal secretions. A new product developed for the control of feline allergic dermatitis was approved by the FDA on 29 March 2023, which demonstrates bioequivalence with Atopica for Cats^®^ [14]. The objective of this study was to investigate the bioavailability of cyclosporine (Atopica for cats) in three different doses in a double-blind trial to assess the rate and amount of dosage that can be absorbed following oral administration. In this study, the blood concentrations of cyclosporine were measured following the oral administration of three different doses of cyclosporine and intravenous administration to healthy cats. These results can illustrate careful dosing guidelines when administering cyclosporine to cats.

## 2. Materials and Methods

### 2.1. Chemical and Reagents

Cyclosporine A (Figure 1) and cyclosporine D (internal standard) (Figure 2) were purchased from First Standard (Tianjin, China) and Sigma-Aldrich (Beijing, China), respectively. HPLC grade acetonitrile, methanol, and formic acid were LC-MS grade and obtained from Fisher Scientific Co. (Fair Lawn NJ, USA). Water was purified through a Milli-Q-synthesis system (Millipore, MA, USA). Ammonium acetate (LC-MS grade) was from Sigma-Aldrich (Beijing, China). Atopic for cats ((Elanco Australasia Pty Ltd., 17 mL/bottle, 100 mg/mL, Greenfield, IN, USA) was provided by Shanghai Hanwei Biomedical Technology.

### 2.2. Standards Solutions

The stock solution of cyclosporine A (0.1 mg/mL) was placed in a brown bottle and stored at −20 °C. The stock solution of cyclosporine D (1 mg/mL) was prepared by dissolving the standard reference into methanol. Working solutions were diluted from the stock solutions using methanol into target concentration and stored at 4 °C.

### 2.3. Method Establishment and Validation

Method establishment used neutered cat blood samples collected at Beijing Yuanda Xinghuo Medicine Technology Co., Ltd. (Beijing China)**.** Cyclosporine A and cyclosporine D were spiked into blank blood samples to establish the calibration curves at the concentrations of 10, 20, 50, 100, 500, 1000, and 2000 ng/mL. Each blood sample was analyzed for the concentration of cyclosporine using ultra-performance liquid chromatography–tandem mass spectrometry (UPLC-MS/MS). The UPLC equipment used in this study consisted of a Thermo vanquish ultra-performance liquid chromatography system coupled with a tandem mass spectrometer, Thermo Quantis, equipped with an electron spray ionization source. All the sample analyses were performed at 4–10 °C using an aqueous mobile phase (solvent A) containing formic acid at a concentration of 0.1% and ammonium acetate at a concentration of 2 mM. The organic mobile phase (solvent B) consists of methanol containing formic acid at a concentration of 0.1% and ammonium acetate at a concentration of 2 mM. The gradient elution program consisted of an initial isocratic separation of 80% aqueous mobile phase for 0.80 min followed by a linear increase in aqueous phase concentration to 100% at 0.85 min and then decreased to 0% at 2.5 min then for a further 3.5 min before returning to initial conditions. The column used was a Waters BEH C18 2.1 × 100 mm analytical column with a flow rate of 0.3 mL/min, the ejection of the volume was 3 µL and the temperature of the column was 60 °C. MS tuning was operated in positive electrospray ionization (ESI^+^) individual solution of cyclosporine A and cyclosporine D together with organic mobile (100%, 0.1 mL/min). The following parameters were used: ionic voltage: 3000 V, sheath gas: 50 Arb, aux gas: 25 Arb, ion transfer tube temp: 320 °C, and vaporizer temp: 330 °C.

### 2.4. Animals

This study was designed in harmony with Good Clinical Practice guidelines, and all protocol were approved by China Agricultural University Animal Welfare Committee (14505-E-21-002). Twenty-four domestic shorthair cats with mean ± SD body weight of 3.40 ± 0.38 kg and age of 2.5 ± 0.5 years were enrolled in this study according to China Agricultural University Animals Welfare. The cats were accommodated in a room with a 12 h light/ 12 h dark cycle, and the room temperature and humidity were set in the range of 20–26 °C and 35–75%, respectively. The animals were provided with food and tap water *ad libitum*. Corticosteroids or antihistamines were not applied in the adaption period. After fourteen days of adaptation period, samples for hematology, serum chemistry, and toxoplasma gondii titers were obtained and evaluated before the study and at the end of the study. Toxoplasma-seropositive cats seem have a protective function against cyclosporine-associated toxoplasmosis as cyclosporine administration did not reactivate quiescent infection and oocyst shedding [15,16,17,18,19]. Cats presenting with clinical signs indicative of bacterial and/or fungal infection or flea or food allergies were excluded from the study with the feline immunodeficiency virus (FIV) or feline leukemia virus (FeLV) positive cats.

### 2.5. Study Design

This study was performed to determine both the bioavailability and the pharmacokinetics of cyclosporine in cats. Animals were randomized to four groups (3/sex/group), namely, intravenous group (3 mg/kg), low oral group (3.5 mg/kg), medium oral group (7 mg/kg), and high oral group (14 mg/kg). Body weight was accessed before initialing the experiments. Each animal in this study was given the drug only once following the animals’ weight. The oral solution (Atopic for cats (ciclosporin oral solution, USP) modified; Elanco Animal Health) was identical and given after fasting food for 12 h. In the intravenous group, samples were obtained at predose,5 min, 15 min, 30 min, 45 min, 1 h, 2 h, 3 h, 4 h, 6 h, 8 h, 12 h, 24 h, 36 h, 48 h, 72 h, and 168 h, and in oral group, samples were achieved at following time points: predose, 15 min, 30 min, 45 min, 1 h, 1.5 h, 2 h, 3 h, 4 h, 6 h, 8 h, 12 h, 24 h, 36 h, 48 h, 72 h, 168 h, and 192 h. At each scheduled time point, 1 mL of blood was collected at the cephalic vein with an Elizabethan collar holding. Then blood sample was transferred into tube flushed with heparinized saline and stored at −80 °C until analysis. During the study, clinical observations and ophthalmic examination results were recorded daily.

### 2.6. Sample Preparation

Frozen whole blood samples were thawed and vortexed, and 95 µL whole blood, 5 µL internal standard (500 ng/mL), and 900 µL methanol were mixed. Then samples were shaken for 3 min and centrifuged for 10 min at 11,100× *g*. After supernatants were filtered through a 0.22 µm microbore cellulose membrane, the samples were collected and bottled to UPLC-MS/MS system for analysis.

### 2.7. Data Analysis

Blood concentrations of cyclosporine were analyzed using the established method, and the pharmacokinetic parameters were calculated via non-compartmental and compartment analysis in WinNonlin software (WinNolin 8.3.4 Certara, Pharsight, Mountain View, CA, USA) and expressed in mean ± standard deviation (SD), and power model and one way ANOVA tests were applied. The difference was significant when the *p*-value was smaller than 0.05.

## 3. Results

The mass spectrum parameters of cyclosporine and the internal standard were: cyclosporine A m/z 1219.9→1202.7 (quantification ion); 1219.9→1184.7 (qualitative ion), cyclosporine D m/z (1235.0→1217.2), respectively. The limit of detection of this assay was 3 ng/mL, and the limit of quantitation was 10 ng/mL. Only concentrations equal to or exceeding 10 ng/mL were covered for the final pharmacokinetic analysis. The chromatograms for the blank matrix and blank matrix spiked with cyclosporine and the internal standard are illustrated in Figure 3 and Figure 4, respectively.

The whole blood concentration–time curves of cyclosporine in cats through intravenous and oral administration are shown in Figure 5. The major pharmacokinetics parameters of cyclosporine in cats using non-compartment analysis are available in Table 1 and Table 2. Following intravenous administration, the AUC_0-t_ was approximately 21,626.06 ± 3534.97 h × ng/mL with C_max_ 2905.19 ± 620.99 ng/mL and t_max_ 0.17 ± 0.19 h. V_d_/F was 6209.35 ± 2224.81 mL/kg. The elimination half-time (t_1/2_) and clearance (CL/F) were 30.3 ± 7.25 h and 139.56 ± 25.19 mL/h/kg, respectively. In the low oral group, the AUC_0–t_ was 3693.34 ± 2579.60 h × ng/mL with C_max_ 778.21 ± 268.90 ng/mL and t_max_ 1.08 ± 0.38 h. V_d_/F was 38,360.27 ± 42,308.51 mL/kg. The elimination half-time (t_1/2_) and clearance (CL/F) were 8.06 ± 4.45 h and 2758.92 ± 1823.53 mL/h/kg, respectively. In the medium oral group, the AUC_0–t_ was 18,658.75 ± 8235.01 h × ng/mL with C_max_ 1392.45 ± 380.67 ng/mL and t_max_ 0.92 ± 0.13 h. V_d_/F was 24,997.17 ± 3121.14 mL/kg. The elimination half-time (t_1/2_) and clearance (CL/F) were 25.82 ± 10.22 h and 920.51 ± 802.73 mL/h/kg, respectively. In the high oral group, the AUC_0–t_ was 13,649.08 ± 7866.6 h × ng/mL with C_max_ 1382.55 ± 570.62 ng/mL and t_max_ 1.42 ± 0.38 h. V_d_/F was 35,292.64 ± 16,953.66 mL/kg. The elimination half-time (t_1/2_) and clearance (CL/F) were 20.12 ± 7.67 h and 1469.43 ± 1132.71 mL/h/kg, respectively. Clearance and the volume of the distribution of cyclosporine were used as the initial values to estimate the parameter of the compartmental model. Secondary parameters are listed in Table 3. Two compartmental models were used to estimate the process of cyclosporine in cats. Following intravenous administration, half time of the central chamber, half time of the peripheral chamber, and half time of elimination were 2.12 ± 1.39 h, 36.12 ± 22.50 h, and 10.18 ± 3.19 h, respectively. In the oral low group, half time of the central chamber, half time of the peripheral chamber, half time of elimination, and half time of absorption were 0.77 ± 0.52 h, 26.85 ± 30.99 h, 1.75 ± 1.11 h, and 1.80 ± 2.50 h, respectively. In the medium oral group, half time of the central chamber, half time of the peripheral chamber, half time of elimination, and half time of absorption were 0.93 ± 0.17 h, 20.53 ± 6.89 h, 4.43 ± 1.55 h, and 0.93 ± 0.17 h, respectively. In the high oral group, half time of the central chamber, half time of the peripheral chamber, half time of elimination, and half time of absorption were 0.96 ± 0.14 h, 31.17 ± 12.51 h, 3.53 ± 0.90 h, and 0.96 ± 0.14 h, respectively. The linear mixed effect model was used to evaluate the linearity in the range of 3.5–14 mg/kg, and the results are provided in Table 4.
F = (AUC_p o_/Dose_p o_)/(AUC_i v_/Dose_i v_) × 100%(1)

The bioavailability was calculated using Formulation (1), and absolute bioavailability values in the low oral group, medium oral group, and high oral group were 14.64%, 36.98%, and 13.53%, respectively. Theoretically, the clearance is a constant. However, when some conditions change to an extreme, the metabolism or elimination is usually saturable due to the drug–enzyme illustrating a nonlinear profile. In this view, clearance will be influenced by the drug concentration. A Michaelis–Menton kinetic model should be used to estimate the V_max_ and K_m_ to calculate clearance [20]. The result indicated that the for oral administration of cyclosporine is not linear in the range of 3.5–14 mg/kg due to the *p*-value being less than 0.01 and the CI range beyond reference 0.839~1.161.

Some scholars have verified that whole blood concentration 2 h after oral administration has a better correlation with AUC_0–12_ than trough concentration [4,21]. As the recommended dose of Atopic in cats is 7 mg/kg/day [13], the AUC0–24 will be more practical in clinic than AUC_0–t_ and AUC_0–∞_. In this study’s four groups, C_4_ (R^2^ = 0.896) (Figure 6) has a higher regression coefficient than C_2_ (R^2^ = 0.707) (Figure 7), which means that it is a more practical and powerful time point to monitor the drug therapy.

## 4. Discussion

Few references have been reported on the pharmacokinetic process in cats. M.L. Mehl evaluated cyclosporine after intravenous and oral administration, which is similar to this study, and the oral administration was 3 mg/kg b.i.d., which showed the value of bioavailability is about 29 and 25% on days 7 and 14, respectively. In our study, the medium oral group demonstrated almost the same result [21]. Whole blood concentrations 2 h after administration were better correlated with AUC than trough concentrations. Yang investigated two cyclosporine oral solutions in cats to demonstrate the two formulations’ bioequivalence [22]. A single oral dosage at 7 mg/kg was given to display a little longer T_max_ than in our study. Lai attempted to enhance the bioavailability of cyclosporine in beagle dogs in a nanoparticle formulation. He demonstrated that nanoparticles had a higher blood concentration than Neoral and a faster elimination rate [23], which held about 178% bioavailability compared to Neoral. He also used a two compartmental model to simulate the process of the drug in dogs. The absorption rate in beagle dogs was longer than that in cats.

Nonlinear pharmacokinetic drugs were those drugs when the kinetics of the drug’s changes were not proportional to the dose changes. The reason for this nonlinearly is that there are one or more processes of the drug that are more complicated than simple first-order kinetics [24]. The area under the blood concentration–time curve by dose normalization is not a constant. In this study, more than half of cyclosporine is not combined, and the free form of cyclosporine is metabolized by the enzyme when following intravenous administration.

Due to its high lipophilic characteristics, cyclosporine is absorbed by passive diffusion, which was affected by P-glycoproteins and the cytochrome [4]. According to the Food and Drug administration Biopharmaceutical Classification Scheme, it belongs to class IV, which has low-solubility and low-permeability properties [25]. Its pharmacokinetic procedure is different in various formulations. As a narrow therapeutic index drug (NTID), trough blood concentration or that 2 h after oral administration were usually measured to monitor the concentration of cyclosporine [21].

Absorption of the traditional formulation, Sandimmune, made with propylene glycol, corn oil, or castor oil, was influenced by many factors, causing its low bioavailability and high variability. However, the ultramicronized counterpart forms microemulsions dissolving in aqueous fluids without concerning the biliary action, enzymes, or small intestinal secretions [2]. W.H. Barr elucidates that the extent of absorption of cyclosporine was affected mostly by jejunum length, physically. The net small intestinal transit time, which is affected by two counter factors, namely CYP 3A4 and P-glycoprotein, is one of the important determinants in bioavailability [25]. The amount of cyclosporine absorbed into the bloodstream depends on a fraction of metabolism in the gut before absorption [26].

Cyclosporine is more susceptible to being accumulated in the skin, liver, and fat than the blood [2]. Niels used DESI and MALDI mass spectrometry imaging technology to demonstrate that cyclosporine did not benefit from tissue washing due to it being lipophilic. The whole imaging showed that cyclosporine was distributed in the whole body and organs with the highest abundance were the pancreas and the liver [27]. The skin holds a higher level of cyclosporine than the blood after being treated for seven days with 14 mg/kg due to its hydrophobic properties [2,4].. However, the concentration in the brain was lower than in the blood due to the P-glycoprotein reducing the amount across the blood–brain barrier [1].

The metabolism of cyclosporine in vivo was different among species. Rats were liable to hepato- and nephrotoxicity with their lower enzyme activity than dogs. The different amounts in P-glycoprotein accounts for the highly individual variation. There are lots of agents that increase or decrease the concentration of cyclosporine in vivo by competitive inhibition or by the induction of enzymes, such as CYP 3A4, P-glycoprotein, and calcium channels. Ketoconazole, which can inhibit both CYP 3A4 and P-glycoprotein enzymes, has been proven to reduce the metabolism of cyclosporine and the dose for keeping the therapy effects. The furanocoumarins in grapefruit juice may inhibit the enzymes to increase the oral bioavailability of many substrates of CYP 3A4 [1,2,25].

The elimination of cyclosporine consists of metabolism in the liver by cytochrome P-450 3A and excretion by bile into feces [28]. A total of 30 metabolites of cyclosporine were found in bile and few were excreted in urine. The hepatic CYP 450 enzyme results showed a highly individual variation [1,2]. Although cyclosporine shows hepatic and intestinal metabolism, the clearance can be affected by renal blood flow fractions. After renal transplant, patients have opportunities to experience the toxicity induced by cyclosporine [29,30].

The most common disorders are always observed gastrointestinally. In human cases, side effects, such as nephrotoxicity, neurotoxicity, and diabetes mellitus, were observed at a high trough plasma of 400–600 ng/mL due to high blood levels and low lymphocyte levels [6], and it was reported that cyclosporine has a more negative weighting change than the control [19], perhaps because of its negative effect on glucose homeostasis [1]. Long-term use of cyclosporine can increase the risk of infection because of its immunosuppressive property [31]. Adverse reactions including vomiting, anorexia, diarrhea, behavioral changes, lethargy, weight loss, hypersalivation, ocular discharge, and sneezing were observed [19,31]. The physicochemical properties of whole blood changed according to the information summary. Mean alkaline phosphatase, calcium, cholesterol, creatinine, glucose, total bilirubin, and urea nitrogen were higher in the ATOPICA for Cats group compared to the control group in the normal ranges. The mean aspartate aminotransferase, magnesium, eosinophil count, and white blood cell count were lower in the ATOPICA for Cats group compared to the control group in the normal ranges [19]. These side effects reduce when the dose decreases or is withdrawn [1]. However, no side effects were observed in this study.

It is believed that precision medicine will be the trend in the future to decrease costs and toxicity through adequate models and data. However, many clinical drugs have monitored the concentration of their level because the relationship between effect and exposure is not clear or has a lag time, there is a narrow gap between therapeutic and toxicity levels, and a high variability [32]. In this way, monitoring the concentration of cyclosporine in the blood is necessary to ensure safety and to adjust the dose to avoid side effects. One common method is monitoring the trough concentration of cyclosporine after oral administration, but there is no guideline on the relationship between the concentration and therapeutic effect. Another one is measuring a 2 h concentration of cyclosporine after oral administration as an approximation of the peak concentration and extrapolating the extent of absorption [21].

## 5. Conclusions

Following oral administration, the absolution bioavailability of Atopic for cats is a little low, and there is high individual variability. The bioavailability of cyclosporine was not dose-dependent, and cyclosporine had a nonlinear pharmacokinetic profile in the range of 3.5–14 mg/kg. In clinical use, C4 could be a powerful predictor for its therapy monitoring. If it was widely used in veterinary medicine, it would have great potential to be a therapeutic treatment after clarifying the beneficial and harmful aspects.

## Figures and Tables

**Figure 1 vetsci-10-00399-f001:**
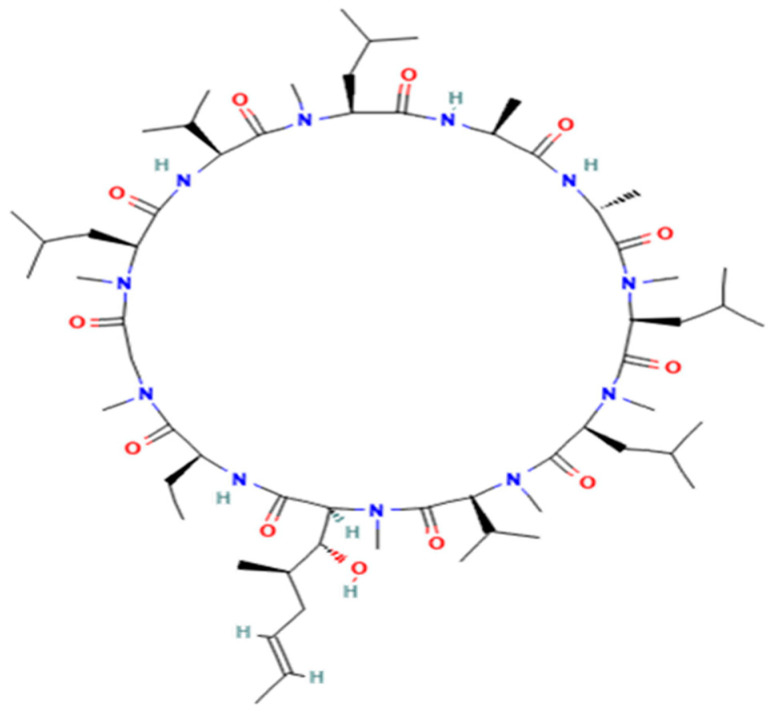
The structure of cyclosporine A.

**Figure 2 vetsci-10-00399-f002:**
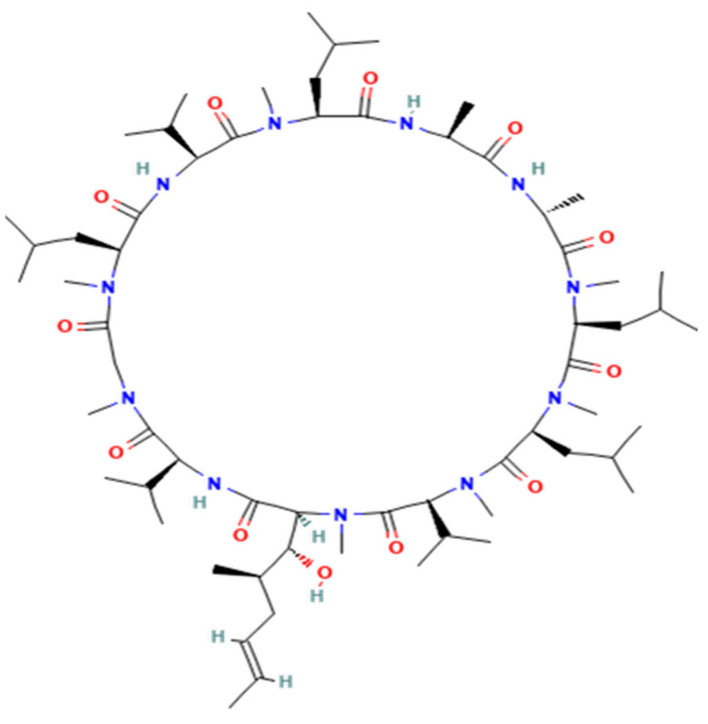
The structure of cyclosporine D (internal standard).

**Figure 3 vetsci-10-00399-f003:**
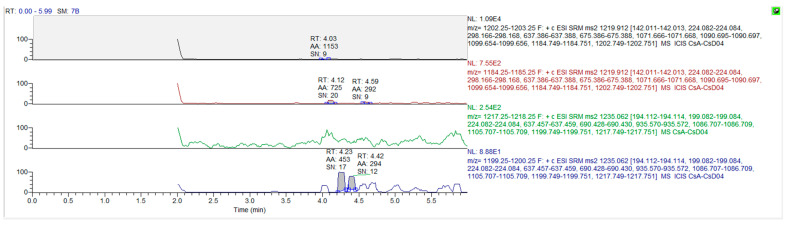
The chromatograms for cyclosporine A and cyclosporine D in blank whole blood.

**Figure 4 vetsci-10-00399-f004:**
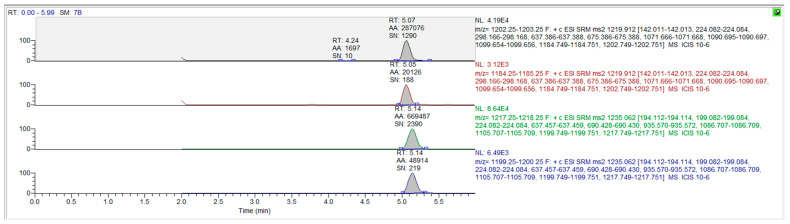
The chromatograms for cyclosporine A and cyclosporine D in blank whole blood spiked drug and internal standard.

**Figure 5 vetsci-10-00399-f005:**
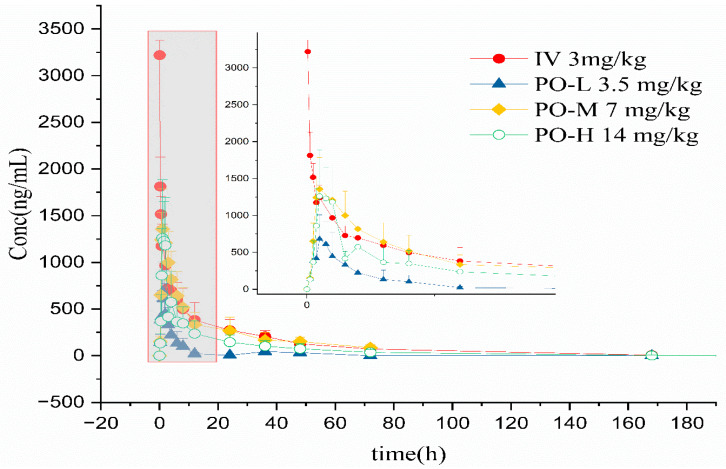
Whole blood concentration–time curves of cyclosporine after intravenous and oral administration.

**Figure 6 vetsci-10-00399-f006:**
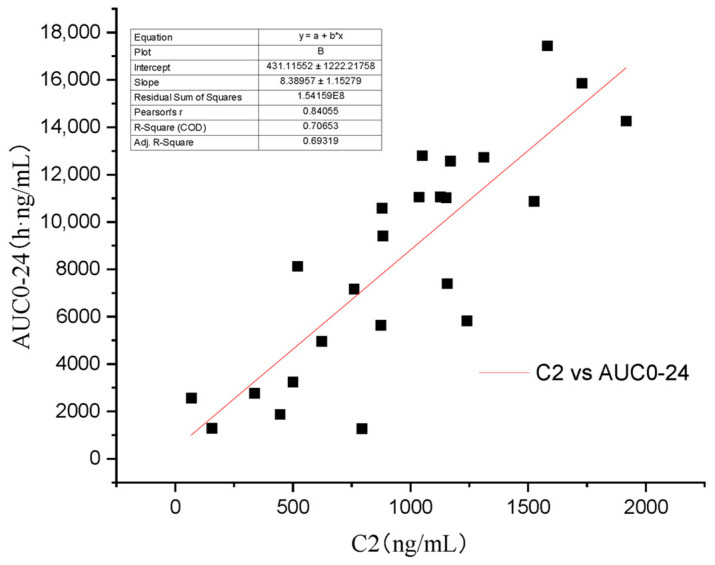
Regression analysis of the relationship between C_4_ and AUC_0–24_.

**Figure 7 vetsci-10-00399-f007:**
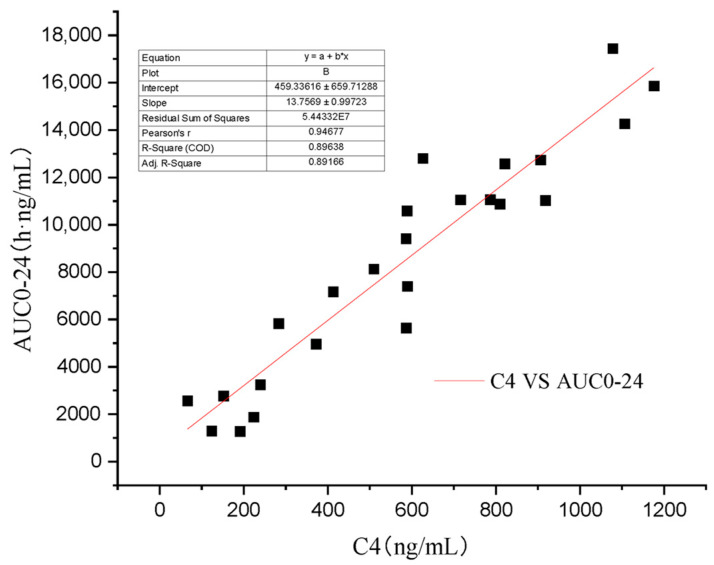
Regression analysis of the relationship between C_2_ and AUC_0–24_.

**Table 1 vetsci-10-00399-t001:** The major pharmacokinetic parameters of cyclosporine A in intravenous group using non-compartment analysis.

Parameters	Intravenous Group
t_max_ (h)	0.17 ± 0.19
C_max_ (μg/mL)	2.91 ± 0.62
t_1/2_ (h)	30.3 ± 7.25
AUC_0–t_ (h’μg/mL)	21.6 ± 3.53
AUC_0–∞_ (h´μg/mL)	22.0 ± 3.46
V_d_ (L/kg)	6.21 ± 2.22
CL (L/h/kg)	0.140 ± 0.0252
MRT_0–t_ (h)	31.9 ± 2.04
AUC_0–4_ (h´μg/mL)	4.12 ± 0.309
AUC_0–24_ (h´μg/mL)	11.0 ± 1.88
AUC/Dose	7.33 ± 1.15
Vss_obs (L)	4930 ± 1230

**Table 2 vetsci-10-00399-t002:** The major pharmacokinetic parameters of cyclosporine A in oral groups using non-compartment analysis.

Parameters	Low Oral Group	Medium Oral Group	High Oral Group
t_max_ (h)	1.08 ± 0.38	0.92 ± 0.13	1.42 ± 0.38
C_max_ (μg/mL)	0.778 ± 0.269	1.39 ± 0.381	1.38 ± 0.571
t_1/2_ (h)	8.06 ± 4.45	25.8 ± 10.2	20.1 ± 7.67
AUC_0–t_ (h´μg/mL)	3.69 ± 2.58	18.7 ± 8.24	13.6 ± 7.87
AUC_0–∞_ (h´μg/mL)	3.74 ± 2.58	21.9 ± 10.6	13.9 ± 7.69
V_d_/F (L/kg)	38.4 ± 42.3	25.0 ± 3.12	35.3 ± 17.0
CL/F (L/h/kg)	2.76 ± 1.82	0.921 ± 0.803	1.47 ± 1.13
MRT_0–t_ (h)	13.2 ± 6.98	18.9 ± 5.47	22.6 ± 8.69
AUC_0–4_ (h´μg/mL)	1.57 ± 0.816	3.97 ± 1.29	3.02 ± 1.14
AUC_0–24_ (h´μg/mL)	2.57 ± 1.63	11.9 ± 4.52	8.13 ± 3.89
AUC_0-∞_/Dose	1.07 ± 0.737	3.13 ± 1.51	0.993 ± 0.549

**Table 3 vetsci-10-00399-t003:** The major pharmacokinetic parameters of cyclosporine A in cats using compartment analysis.

Parameters	Intravenous Group	Low Oral Group	Medium Oral Group	High Oral Group
A	1170 ± 127	−312,000 ± 542,000	79,200,000 ± 144,000,000	−0.38 ± 0.36
B	479 ± 292	216 ± 451	557 ± 201	2220 ± 919
α	0.45 ± 0.27	1.17 ± 1.26	0.75 ± 0.08	2.49 ± 0.02
β	0.02 ± 0.01	0.42 ± 0.14	0.04 ± 0.02	0.24 ± 0.07
t_1/2 α_ (h)	2.12 ± 1.39	1.07 ± 0.69	0.93 ± 0.08	0.28 ± 0.00
t_1/2 β_ (h)	36.1 ± 22.5	1.81 ± 0.61	21.3 ± 5.68	3.06 ± 0.80
t_1/2 e_ (h)	10.2 ± 3.19	1.16 ± 1.27	4.52 ± 1.46	3.06 ± 0.80
K_12_	0.23 ± 0.11	0.00 ± 0.00	0.45 ± 0.02	0.00 ± 0.00
K_21_	0.16 ± 0.16	0.43 ± 0.14	0.16 ± 0.02	2.49 ± 0.02
K_e_	0.07 ± 0.02	1.16 ± 1.27	0.18 ± 0.09	0.24 ± 0.07
t_1/2 Ka_ (h)		0.93 ± 0.28	0.93 ± 0.08	0.51 ± 0.10

**Table 4 vetsci-10-00399-t004:** The result of linear property for cyclosporine using linear mixed effect model.

Models	Estimate	F-Stat	*p*-Value	Lower CI (95%)	Upper CI (95%)
Power model	0.96			0.24	1.69
One Way ANOVA		10.98	0.001		

## Data Availability

No new data were created or analyzed in this study. Data sharing is not applicable to this article.

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
