# Peer review of "The Pharmacokinetic and Absolute Bioavailability of Cyclosporine (Atopica for Cats®) in Cats"

_vetsci, 2023, doi:10.3390/vetsci10060399_

Round 1

Reviewer 1 Report

This is a very well-written and well-described manuscript summarizing interesting and important findings of a drug that is increasingly commonly used in feline practice.

Reviewer 2 Report

The manuscript contributes to the cyclosporine drug disposition literature for cats, but has some correctable and some non-correctable difficiencies:

Correctable:

1. Use of significant digits: Based on the accuracy and precision of the analytical methods, and the accuracy of the timing of blood sampling, the calculated PK values should report no more than three significant digits (12300, 1230, 123., 12.3, 1.23, 0.123, 0.0123, 0.00123, etc.). 

2. For PK values that are known either to be categorical (eg, Tmax) or non-normally distributed by definition, the use of mean +/- standard deviation is incorrect. The authors should review the use of summary statistics for such data.

3. Section 2.7 states that no statistical tests were performed, yet p-values were listed in Table 3. 

Non-correctable:

1. Statistical assessment of oral bioavailability cannot be not correctly calculated unless the oral vs. IV comparisons are within animal. In this experimental design, the only thing that can be done is to statistically compare the AUC (through infinity) in the various groups in a completely randomized design. While it may be clear that there is a statistical difference in the AUC, the bioavailability cannot be determined since systemic clearance across the different treatment groups cannot be assumed to be identical. 

2. Similar comments can be made for the C2 - AUC and C4 - AUC discussion. Furthermore, any comparisons between the two should use a statistically valid approach. This reviewer does not believe that an R2 of 0.896 is significantly different from 0.707. Unless there is a statistical difference in the two, any reference to a different utility is premature. 

1. English grammar and punctuation throughout the manuscript requires substantial editing and correcting. This would benefit from a native English-speaking editor.

Reviewer 3 Report

Thank you for the opportunity for reviewing.

The authors investigated the PK of Ciclosporin (Atopica) in cats administered IV or PO at three different dose levels. F ranged from approximately 13 - 37%. Dose nonproportionality / nonlinearity was observed. The blood concentration at 4h correlated best with exposure. Overall, this is an interesting paper with practical and clinical relevance, however, there are some major issues that need to be addressed before approving and accepting the paper.

GENERAL COMMENTS

The English used in the manuscript contains many grammatical mistakes and hampers the readability and message of the paper. I suggest to let an expert on the language proofread, correct and/or amend the text. 

INTRODUCTION

The description of the mode of action of ciclosporin is not clear and info is sometimes repeated. Consider rephrasing.

Lines 55 - 71 are also not very clear. I feel this is a bit of a random order of information with no structure. Consider rephrasing.

Missing: What is the difference between Atopica and the other mentioned formulations? Are there other veterinary approved products?

M&M

Fig 1 and 2: include the reference. Is it also possible to use a white background instead of grey?

How were the doses for cats decided? The dose on the label of Atopica is 5mg/kg for dogs, which is not tested here? 

RESULTS

For IV, the Vd and Cl values should not be /F as IV has F of 100% by definition. Also the results of the parameters displayed for the oral routes should be corrected for F

Which compartmental model is used?

It is not clear to me how the nonlinearity / dose nonproportionality was assessed. Please elaborate. How do you explain the F is larger in the medium group then in the high group?

DISCUSSION

The discussion is in my opinion poorly written and provides information randomly and unstructured. There is also almost no comparison made with other studies investigating PK in cats.

What is the clinical relevance if the results of this study?

SPECIFIC COMMENTS

INTRODUCTION

Line 28-30: is this the IUPAC name? It does not correspond to what I find on the internet.

SandImmune, Neoral: are these products for human medicine? Specify in the texts 

M&M

Line 101: what does "neutral" sample mean?

Line 109: typo "spray"

line 126: include standard deviation (SD) for weights. What is the mean age + SD?

line 131: "adaptation period"

Line 134: please elaborate on the protective effect of Toxoplasma against ciclosporin

Line 151: which anticoagulant? The sentence is also not proper English.

Line 162: In the abstract it is stated that also compartmental modelling was done?

Figure 3 and 4 are not very clear and particularly relevant. Consider deleting.

Line 181: "noncompartmental analysis", not model, as you assume using NCA  no model structure / compartments

Table 1: also include AUC/D, as this makes comparison between doses easier

Line 233: Why AUC24h better?

Fig 6 and 7: no units are displayed. Why are not other concentrations tested? Are the R2 coefficients statistically significantly different from each other? What is the relationship between PK and clinical response? Please elaborate.

Please consult English expert and revise the text substantially.

Round 2

Reviewer 3 Report

Dear Authors,

Thank you very much for revising the paper.

The uploaded latest version does not contain the line numbers and the comments on the right of the pdf are cut off. Would it be possible to upload a new version?

Additionally, I have some other remarks:

- The English has been improved. However, throughout the text, there are still some moderate editing / spelling / grammar mistakes.

- Drug formulation used: is this a human formulation? Veterinary? Commercially available? Specify in the text.

- AUC/D --> Which AUC, specify

- Regarding the assessment of linearity, thank you for amending and providing the tests. Is it possible to give a bit of background on these tests in order for the reading audience to understand.

- Regarding the compartmental modeling: please provide more background on the model: which model? What is the structure (consider including a figure)?  What are the CV% for each parameter? Etc. This is crucial for reproducibility.

- The discussion is still very unstructured and not clearly written. It is not clear in several paragraphs whether the authors are talking about human or veterinary medicine and there is no clear link between the paragraphs.

- Missing from discussion: relevance of C4 for the clinical response

Still a moderate revision needed.
